# Is There Mode Collapse? A Case Study on Face Generation and Its Black-box Calibration

## Abstract

Generative adversarial networks (GANs) nowadays are capable of producing images of incredible realism. One concern raised is whether the state-of-the-art GAN's learned distribution still suffers from mode collapse. Existing evaluation metrics for image synthesis focus on low-level perceptual quality. Diversity tests of samples from GANs are usually conducted qualitatively on a small scale. In this work, we devise a set of statistical tools, that are broadly applicable to quantitatively measuring the mode collapse of GANs. Strikingly, we consistently observe strong mode collapse on several state-of-the-art GANs using our toolset. We analyze possible causes, and for the first time present two simple yet effective "black-box" methods to calibrate the GAN learned distribution, without accessing either model parameters or the original training data.

## 1 Introduction

Generative adversarial networks (GANs) (Goodfellow et al., 2014) have demonstrated unprecedented power for various image generation tasks. However, GANs have also been suffering from generation bias and/or loss of diversity. The underlying reasons could be compound, ranging from the data imbalance to the training difficulty of GANs, and more:

- First of all, the training data for GANs, especially for the typical unconditional/unsupervised generation tasks (Karras et al., 2017; 2018), might possess various subject or attribute imbalances. As a result, GANs trained with them might be further biased towards the denser areas, similarly to the classifier bias towards the majority class in imbalanced classification.

- More intrinsically, even when the training dataset "looks" balanced, training GANs is notoriously more unstable (sometimes even uncontrollable) than training classifiers, potentially constituting another source of mode collapse. One most common hurdle of GANs is the loss of diversity due to mode collapse (Goodfellow, 2016), wherein the generator concentrates too large a probability mass on a few modes of the true distribution. Another widely reported issue, known as co-variate shift (Santurkar et al., 2017), could be viewed as a nuanced version of mode collapse.

This paper seeks to explore: do the state-of-the-art GANs still suffer from mode collapse? Can we have a toolkit to detect that? And if the mode collapse happens, is there any "easy and quick" remedy for calibrating the GAN's learned distribution to alleviate the mode collapse?

**Evaluation of Mode Collapse** There are several popular metrics for GAN evaluation, *e.g.* Inception Score (IS) (Salimans et al., 2016), Fréchet Inception Distance (FID) (Heusel et al., 2017), MODE (Che et al., 2016) and birthday paradox based diversity test (Arora & Zhang, 2017). IS, FID and MODE score takes both visual fidelity and diversity into account. Birthday paradox based diversity test gives a rough estimation of support size under the assumption of uniform sampling. Recently, a classification-based metric (Santurkar et al., 2017) was proposed for a quantitative assessment of the mode distribution learned by GANs. However, their approach hinge on a classifier trained on the original (balanced) GAN training set, with class labels known, available and well-defined (*e.g.*, object classes in CIFAR-10, or face gender in CelebA), making it non-straightforward to extend to data subjects where classes are hard to be defined, and/or are not enumerable (e.g, open set problems).

To tackle this problem, we propose a hypothesis test method by analyzing the clustering pattern of samples. We exploit a statistical tool from spatial analysis, called Ripley's K function, to quantitatively

measure the mode collapse. We demonstrate the application of our tool set in analyzing the bias in *unconditional face image generation*: a popular benchmark task nowadays for GANs, yet remaining rather unclear how to measure its mode collapse using existing tools since every generated identity is expected to be new. The study of face identity generation bias has profound practical values for understanding facial privacy (Filipovych et al., 2011) and fairness (Holstein et al., 2018). Using our tools, we find the mode collapse still a prevailing problem in state-of-the-art face generation GANs (Karras et al., 2018; 2017), and further analyze several possible causes.

**Calibration Approaches on GAN**   Many approaches have been proposed to alleviate mode collapse problem, ranging from better optimization objectives (Arjovsky et al., 2017; Mao et al., 2017), to specialized builing blocks (Durugkar et al., 2016; Ghosh et al., 2018; Liu & Tuzel, 2016). However, they require either tedious (re-)training, or at least the access to training data, as well as to model parameters: we refer to the existing methods as *white-box* approaches.

In contrast, we are interested in an almost unexplored aspect: assuming some generation bias is known, how can be calibrate the GAN, without accessing either the training data or the current model parameters? Such *black-box* calibration is desirable due to many practical demands: the training data might be protected or no longer available; the GAN model might be provided as a black box and cannot be altered (*e.g.*, as APIs); or we simply want to adjust the generated distribution of any GAN with minimized re-training efforts. For the first time, we explore two "black-box" approaches to calibrate the GAN learned distribution, *i.e.*, latent space reshaping via Gaussian mixture models, and importance sampling. They are observed to alleviate the mode collapse without re-touching training data, nor even needing any access to model parameters.

## 2   RELATED WORKS

### 2.1   EVALUATION METRICS OF MODE COLLAPSE IN GANS

GAN models are often observed to suffer from the mode collapse problem (Salimans et al., 2016); (Sutskever et al., 2015), where only small modes subsets of distribution are characterized by the generator. The problem is especially prevalent for high-dimensional data, *e.g.* face image generation, where the training samples are low-density w.r.t. the high-dimensional feature space.

Salimans et al. (2016) presented the popular metric of Inception Score (IS) to measure the individual sample quality. IS does not directly reflect the population-level generation quality, *e.g.*, the overfitting and loss of diversity. It also requires pre-trained perceptual models on ImageNet or other specific datasets (Barratt & Sharma, 2018). Heusel et al. (2017) propose the Fréchet Inception Distance (FID), which models the distribution of image features as multivariate Gaussian distribution and computes the distance between the distribution of real images and the distribution of fakes images. Unlike IS, FID can detect intra-class mode dropping. However, the multivariate Gaussian distribution assumption hardly holds very well on real images, and low FID score cannot rule out the possibility of the generator's simply copying the training data. Besides the two most popular metrics, (Che et al., 2016) develop an assessment for both visual quality and variety of samples, known as the MODE score and later shown to be similar to IS (Zhou et al., 2017). (Arora et al., 2018) and (Arora & Zhang, 2017) proposed a test based upon the birthday paradox for estimating the support size of the generated distribution. Although the test can detect severe cases of mode collapse, it falls short in measuring how well a generator captures the true data distribution. It also heavily relies on human annotation, making it challenging to scale up to larger-scale evaluation.

(Santurkar et al., 2017) took a classification-based perspective and view loss of diversity as a form of covariate shift. As we discussed above, their approach cannot be straightforwardly extended to data subjects without pre-known and closed-set class definition, in addition to the need of training an extra classifier on the original labeled training set.

### 2.2   MODEL CALIBRATION APPROACHES OF GANS

There are many efforts to address the mode collapse problem in GANs. Some focus on discriminators by introducing different divergence metrics (Metz et al., 2016) and optimization losses (Arjovsky et al., 2017; Mao et al., 2017). The minibatch discrimination scheme allows the discriminator

to discriminate between whole mini-batches of samples instead of between individual samples. (Durugkar et al., 2016) adopted multiple discriminators to alleviate mode collapse. ModeGAN (Che et al., 2016) and VEEGAN (Srivastava et al., 2017) enforce the bijection mapping between the input noise vectors and generated images with additional encoder networks. Multiple generators (Ghosh et al., 2018) and weight-sharing generators (Liu & Tuzel, 2016) are developed to capture more modes of the distribution. However, these approaches are designed to easily calibrating trained GANs.

A handful of existing works attempt to combine GANs with sampling methods to improve generation quality. (Turner et al., 2018) introduced the Metropolis-Hastings generative adversarial network (MH-GAN). The MH-GAN uses the learned discriminator from GAN training to build a wrapper for the generator for improved sampling, at the generation inference stage. With a perfect discriminator, the wrapped generator can sample from the true distribution exactly even with a deficient generator. (Azadi et al., 2018) proposed discriminator rejection sampling (DRS) for GANs, which performs rejection sampling on the outputs of the generator by using the probabilities given by the discriminator, to approximately correct errors in the generator's distribution. Yet still, these approaches are *white-box* calibration since both require access to trained discriminators (which might be even less available/accessible than the generator after a GAN is trained).

## 3 METHOD

We intend to study the bias of the most representative features of the generated faces, *i.e.*the face identity distribution, since almost all face attributes can be derived based on this representations. To detect face identity collapse, we are aiming to detect high-density regions in features space caused by any possible attribute non-diversified. Or, if being slightly imprecise in terms,(Santurkar et al., 2017) examined the marginalized distribution through some discrete categorical attributes' lens, while ours looks at the joint distribution of all possible attributes in the continuous feature space holistically.

---

**Algorithm 1** Identity Clustering Pattern Analysis via Sampling and Neighboring Function $\mathcal{N}$

---

> $\triangleright$ Given a pre-trained generator $G$, an identity descriptor $f_{id}$, a random distribution $N(0, \Sigma)$, a neighbor distance threshold $d_0$ and a face embedding space distance range $[d_b, d_e]_{d_s}$ ($d_s$: step size)

$\triangleright \mathcal{S} \leftarrow \{I_1^{\mathcal{S}}, \cdots, I_m^{\mathcal{S}}\}$            // Randomly sampled $m$ face images

**for each** $I_i^{\mathcal{S}} \in \mathcal{S}$ **do**       // Count neighbors within $d_0$ distance for each sampled $I_i^{\mathcal{S}}$

     $\triangleright N_{I_i^{\mathcal{S}}} \leftarrow \mathcal{N}(I_i^{\mathcal{S}}, \mathcal{S}\backslash I_i^{\mathcal{S}}, d_0)$

$\triangleright \mathcal{R}_{obs} \leftarrow \{\tilde{I}_1^{\mathcal{S}}, \cdots, \tilde{I}_p^{\mathcal{S}}\}$

         // Find the region for observation by selecting the top $p$ face images in $\mathcal{S}$ with largest $N_{I_i^{\mathcal{S}}}$

$\triangleright \mathcal{R}_{ref} \leftarrow \{\hat{I}_1^{\mathcal{S}}, \cdots, \hat{I}_q^{\mathcal{S}}\}$     // Find the region for reference by randomly selecting $q$ face images from $\mathcal{S}$

$\triangleright \mathcal{T} \leftarrow \{I_1^{\mathcal{T}}, \cdots, I_M^{\mathcal{T}}\}$           // Randomly sampled $M$ face images ($M \gg m$)

**for each** $d$ **in** $[d_b, d_e]_{d_s}$ **do**

     **for each** $\tilde{I}_i^{\mathcal{S}} \in \mathcal{R}_{obs}$ **do**       // Count neighbors within $d$ distance for each $\tilde{I}_i^{\mathcal{S}}$ in $\mathcal{R}_{obs}$

         $\triangleright N_{\tilde{I}_i^{\mathcal{S}}}^d \leftarrow \mathcal{N}(\tilde{I}_i^{\mathcal{S}}, \mathcal{T}, d)$

     **for each** $\hat{I}_i^{\mathcal{S}} \in \mathcal{R}_{ref}$ **do**       // Count neighbors within $d$ distance for each $\hat{I}_i^{\mathcal{S}}$ in $\mathcal{R}_{ref}$

         $\triangleright N_{\hat{I}_i^{\mathcal{S}}}^d \leftarrow \mathcal{N}(\hat{I}_i^{\mathcal{S}}, \mathcal{T}, d)$

$\triangleright$ Compute the pointwise confidence regions of $[N_{\hat{I}_i^{\mathcal{S}}}^d |_{1-\frac{\alpha}{2}}, N_{\hat{I}_i^{\mathcal{S}}}^d |_{\frac{\alpha}{2}}]$ for each $d \in [d_b, d_e]_{d_s}$, at confidence level of $\alpha$ (default 0.05). The intervals between the upper and lower confidence bounds for all samples in $\mathcal{R}_{ref}$ define the confidence band (Eubank & Speckman, 1993).

$\triangleright$ Reject the hypothesis that the clustering pattern of $R_{obs}$ is the same as that of $R_{ref}$, if the curve of $N_{\tilde{I}_i^{\mathcal{S}}}^d$ falls outside of the confidence band.

---

Given an unconditional face generator $G$ and an identity descriptor $f_{id}$, we sample images $I = G(z)$ using a random distribution $z \sim N(0, \Sigma)$. The unit vector $f_{id}(I)$ describes the identity feature in the face embedding space. The normalized cosine distance between image $I_0$ and $I_1$ is defined as:

$$d(I_0, I_1) = \frac{1}{\pi} cos^{-1}(< f_{id}(I_0), f_{id}(I_1) >) \tag{1}$$

For a given anchor face image $I_0$, a distance threshold $d_0$ and a collection of randomly sampled face images $\mathcal{S}$, the neighboring function $\mathcal{N}(I_0, \mathcal{S}, d_0)$ is defined to compute the number of neighbors within $d_0$ distance of $I_0$, among all images in $\mathcal{S}$:

$$\mathcal{N}(I_0, \mathcal{S}, d_0) = \sum_{I \in \mathcal{S}} \frac{1}{2}(1 + sgn(d_0 - d(I_0, I))) \tag{2}$$

We refer to the tool of Ripley's K function (Dixon, 2014), a spatial analysis method used to describe point patterns over a given area of interest. Ripley's K function can be used to determine if the points of interest appears to be dispersed, clustered, or randomly distributed throughout the area. Our defined neighboring function $\mathcal{N}(I_0, \mathcal{S}, d_0)$ serves as a surrogate of the Ripley's K function $K(d)$.

**Hypothesis Testing**  Given an observed high-identity-density region $\mathcal{R}_{obs}$ and a reference region $\mathcal{R}_{ref}$, we want to test the hypothesis that the clustering pattern of $\mathcal{R}_{obs}$ is the same as $\mathcal{R}_{ref}$. We use $\mathcal{N}$ to get the clustering pattern for the anchor images in $\mathcal{R}_{obs}$ and $\mathcal{R}_{ref}$ respectively. We can reject the hypothesis if the clustering pattern of $\mathcal{R}_{obs}$ is significantly different from $\mathcal{R}_{ref}$. The detailed algorithm is outlined in Algorithm 1.

## 4 EMPIRICAL STUDY AND ANALYSIS

We choose two state-of-the-art GANs: PGGAN (Karras et al., 2017) and StyleGAN (Karras et al., 2018), as our model subjects of study. Both are known to be able to produce high-resolution, realistic and diverse images. We find that the observations below drawn from the two models also generalize to a few other GAN models. We choose the CelebAHQ benchmark (Karras et al., 2017) and FFHQ benchmark (Karras et al., 2018) as our data subject of study. Both benchmarks are composed of diverse and realistic face images. All images are $1024 \times 1024$ resolution unless otherwise specified.

We use ensemble model of InsightFace (Deng et al., 2019b; Guo et al., 2018; Deng et al., 2018; 2019a), FaceNet (Schroff et al., 2015) and CosFace (Wang et al., 2018) as $f_{id}$ to serve as the face identity descriptor. We emphasize that the due diligence of "sanity check" has been performed on those classifiers, *e.g.*, their face recognition results are manually inspected one-by-one and confirmed to be highly reliable on the generated images. $q$ ($|\mathcal{R}_{ref}|$) is set to be 1000. We empirically set $d_b$, $d_e$ and $d_s$ are set to be 0.1, 0.5 and 0.01 respectively.

### 4.1 OBSERVATION OF THE MODE COLLAPSE

**Mode Collapse Analysis**  For both StyleGAN and PGGAN, despite of the observed diversity and high quality of their generated images, we empirically find some high-density regions in both learned distributions. Figure 1 shows that the clustering pattern of $\mathcal{R}_{obs}$ is significantly different from that of $\mathcal{R}_{ref}$, showing that even the learned distributions of two currently best models have strong dense regions towards some specific identities. For simplicity, our study target is the *worst-case* dense mode, *i.e.* the identity with the largest number of neighbors within a given distance threshold.

**Consistency of the Dense Mode**  The dense region $R_{obs}$ is obtained by selecting the top $p$ images in $\mathcal{S}$ with the largest number of neighbors. In order to test the consistency of the *worst-case* dense mode $I_m$ against sampling, we visualize the $I_m$ w.r.t. different size of $\mathcal{S}$ in Figure 2. We consistently observe roughly the same identity as the sampling size increases. $I_m$ can be reliably obtained even when $|\mathcal{S}| = 1k$. The consistency of $I_m$ demonstrate that the support size of $I_m$ is unnegligible.

### 4.2 EMPIRICAL STUDY OF THE CAUSE OF MODE COLLAPSE

We hypothesize multiple factors that may potentially lead to the observed dense mode of face identity. We perform additional experiments, aiming to validate one by one: unfortunately, **none of them** was observed to reduce the observed mode collapse. That implies the existence of some more intrinsic reason for the mode collapse in GAN, which we leave for future exploration.

**Imbalance of Training Data?**  CelebAHQ is a highly imbalanced dataset: among its $30,000$ high-resolution face images of $6,217$ different celebrities, the largest identity class has $28$ images and the smallest one has only $1$. Would a balanced dataset alleviate the mode collapse?

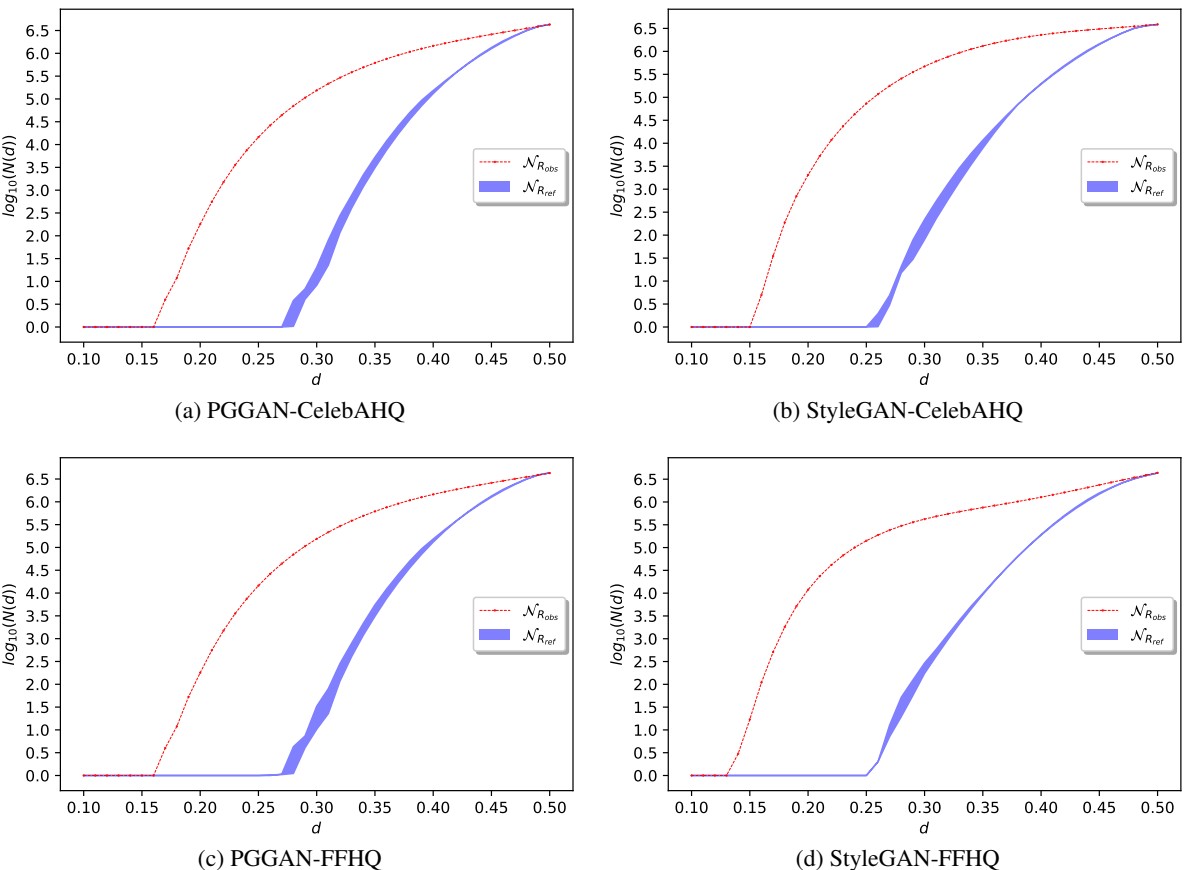

Figure 1: Identity clustering pattern analysis on StyleGAN and PGGAN, both trained on CelebAHQ. The blue region is a confidence band formed by the pointwise intervals between the upper and lower confidence bounds for all identities in $\mathcal{R}_{ref}$. The red curve is the neighboring function curve for identity in $\mathcal{R}_{obs}$, the *worst-case* dense mode. We empirically set $m$ ($|\mathcal{S}|$) to be $100,000$ and $M$ ($|\mathcal{T}|$) to be $10,000,000$. To study the *worst-case* dense mode, $p$ ($|\mathcal{R}_{obs}|$) is set to be 1.

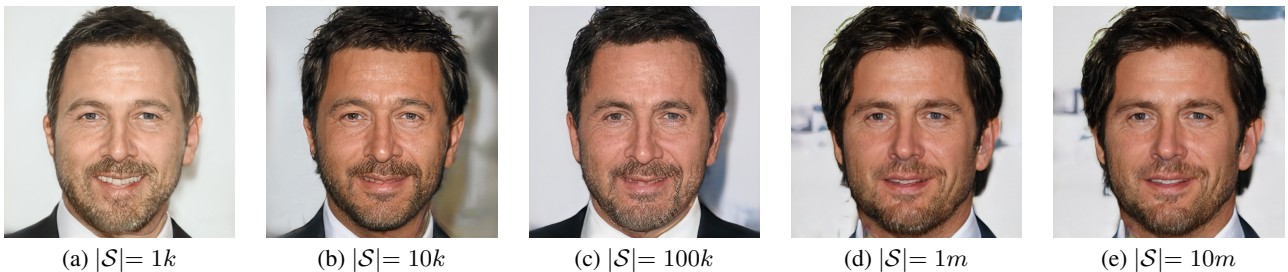

(a) $|\mathcal{S}| = 1k$     (b) $|\mathcal{S}| = 10k$     (c) $|\mathcal{S}| = 100k$     (d) $|\mathcal{S}| = 1m$     (e) $|\mathcal{S}| = 10m$

Figure 2: Visualization of the *worst-case* dense mode $I_m$ w.r.t. different size of the $\mathcal{S}$. $\mathcal{S}$ is a collection of randomly sampled images.

We turn to the Flickr-Faces-HQ Dataset (FFHQ), a high-quality human face dataset created in (Karras et al., 2018), consisting of $70,000$ high-resolution face images, without repeated identities (we manually examined the dataset to ensure so. It is thus "balanced" in terms of identity, in the sense that each identity class has one sample. We train StyleGAN on FFHQ: somehow surprisingly, the mode collapse persists and seems no less than StyleGAN on CelebAHQ, as shown in Figure 1c and 1d.

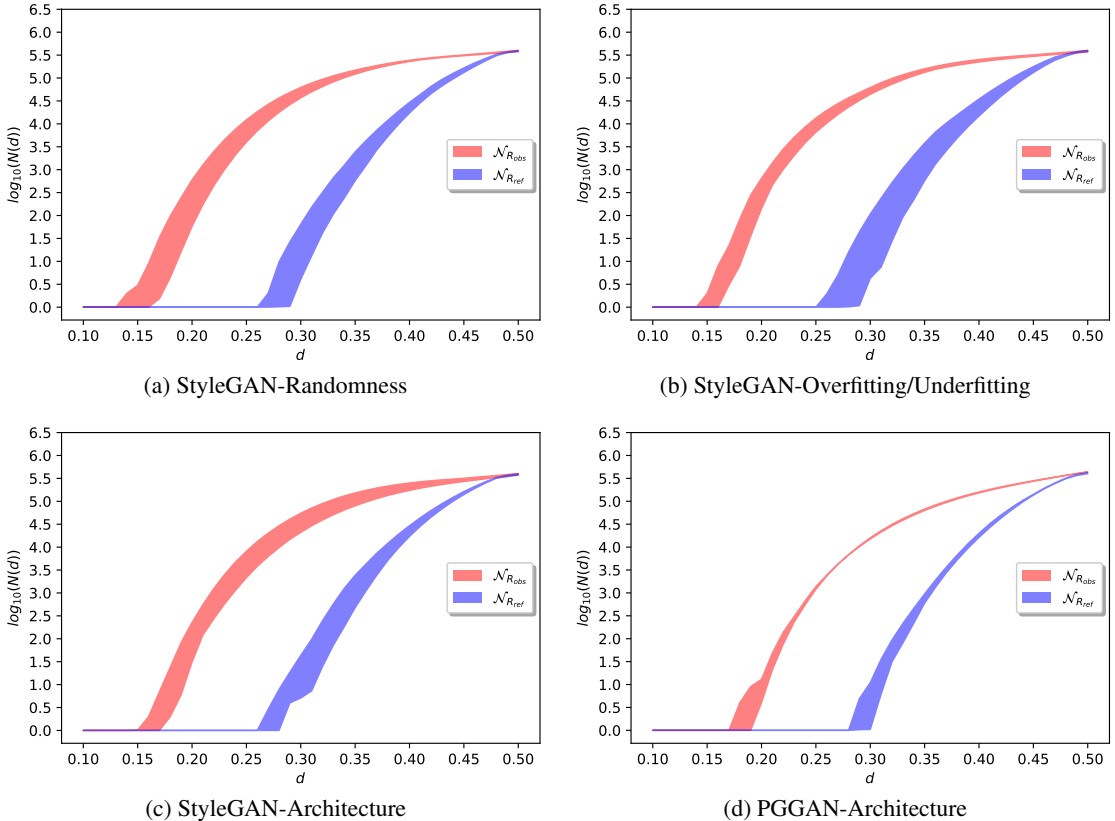

Figure 3: Empirical study on possible causes of the mode collapse. The shaded areas denote the variances of neighboring statistics for different experiments (caused by re-initialization/training; running different iterations; and varying architectures: see the texts for details). We empirically set $m$ ($|\mathcal{S}|$) to be $100,000$ and $M$ ($|\mathcal{T}|$) to be $1,000,000$. To study the *worst-case* dense mode, $p$ ($|\mathcal{R}_{obs}|$) and is set to be 1.

**Randomness during Initialization/Optimization?**    We repeat training StyleGAN on CelebAHQ ($128 \times 128$) for 10 times. The experimental results are shown in Figure 3a, with the shaded areas denoting the variances. Despite the variance for the neighboring function curves plotted for repeated experiments, a large gap between the curves of $R_{obs}$ and $R_{ref}$ can be consistently observed.

**Unfitting/Overfitting in Training?**    We train StyleGAN on CelebAHQ ($128 \times 128$) again, and store model checkpoints at iteration 7707 (FID = 7.67, same hereinafter), 8307 (7.02), 8908 (6.89), 9508 (6.63), 10108 (6.41), and 12000 (6.32). We plot their corresponding neighboring function curves in Figure 3b. Similarly, despite the variances, the identity mode collapse persists due to the consistent large gap between $R_{obs}$ and $R_{ref}$ curves.

**Model Architecture Differences?**    Both StyleGAN and PGGAN progressively grow their architectures that can generate images of different resolutions: 128, 256, 512 and 1024. Utilizing this property, we train StyleGAN and PGGAN on CelebAHQ-128, CelebAHQ-256, CelebAHQ-512 and CelebAHQ-1024 respectively, and plot the neighboring function curves correspondingly. According to Figures 3c and 3d, varying the architectures does not eliminate the mode collapse either.

## 5 BLACK-BOX CALIBRATION APPROACHES

Given a pre-trained generator $G$ and target dense mode for alleviation, the goals of calibration are three-fold: (1) the density of the mode is maximally alleviated; (2) the diversity and quality of the

generated images (measured by FID) are minimally sacrificed; and (3) the calibration is black-box, which does not require access to training data or model parameters.

We propose two calibration approaches: reshaping latent space via Gaussian mixture models and importance sampling. They operate on the latent codes, and require no modification of the trained model, nor even any access to the model parameters or training data, making them "black-box".

Both approaches are evaluated with StyleGAN trained on CelebAHQ-128. For simplicity, we only target to eliminating the *worst-case* dense mode $I_m$, *i.e.* the identity with the largest number of neighbors within a specified distance threshold.

## 5.1 Reshaping Latent Space via Gaussian Mixture Models

Since we consistently observe close neighbors to $I_m$, when interpolating near $I_m$, we hypothesize that the latent codes of a dense mode $I_m$ lay on a smooth manifold. Based on this assumption, we attempt to re-shape that into a Gaussian mixture.

### 5.1.1 Method Description

The original latent space distribution $\phi(z; \theta_0)$ can be approximated with a mixture of Gaussian distributions $\sum_{i=1}^{K} w_i \phi(z; \theta_i)$. We randomly sample $N$ latent code and use $K$-means to explore $\theta_i = (\mu_i, \sigma_i)$. We denote $p(I_m)$ as the probability of sampling the *worst-case* dense mode $I_m$.
$p(I_m) = \int p(I_m|z)\phi(z; \theta_0)dz = \sum_{i=1}^{K} w_i \int p(I_m|z)\phi_i(z; \theta_i)dz$. If $p(I_m|\theta_i)$ is large, we reduce $w_i$ to make the overall $p(I_m)$ small. $p(I_m|\theta_i)$ is estimated by the number of neighbors within $d_0$ distance to $I_m$ in cluster $\mathcal{C}_i$, *i.e.* $\mathcal{N}(I_m, \mathcal{C}_i, d_0)$.

### 5.1.2 Experiments

Starting from a StyleGAN model $\mathcal{M}$ pre-trained on CelebAHQ-128, we aim at alleviating the collapse on the *worst-case* dense mode $I_m$ in $\mathcal{R}_{obs}$ with the largest number of neighbors. We reshape the latent space of $\mathcal{M}$ via Gaussian mixture models to get the new model $\mathcal{M}'$. We get the new *worst-case* dense mode $I'_m$ in the new region $\mathcal{R}'_{obs}$ with the largest number of neighbors. We next randomly sample $10^6$ images from the original Gaussian distribution and new GMM distribution, to form $\mathcal{T}$ and $\mathcal{T}'$ respectively. We then plot the neighboring function curves for $I_m$ in $\mathcal{T}$ and $\mathcal{T}'$, and $I'_m$ in $\mathcal{T}$ and $\mathcal{T}'$ respectively. We expect the reshaping latent space via Gaussian mixture models to alleviate the worst-case dense mode with the minimal sacrifice of generated image quality and diversity.

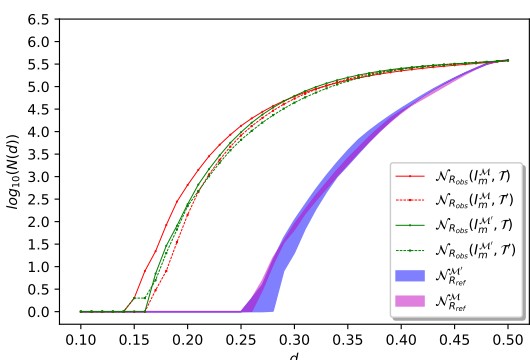

Figure 4: Identity clustering pattern analysis of StyleGAN on CelebA, before/after latent space reshaping.

As shown in Figure 4, the latent space reshaping could suppress the clustering of $I_m$ (indicated by a large gap between the two red curves) without intensifying the clustering of $I'_m$ (indicated by a little gap between the two green curves), resulting in a reduction of mode collapse on $I_m$. Such an alleviation is achieved with an unnoticeable degradation of generation quality, with FID increasing from 5.93 ($\mathcal{M}$) to 5.95 ($\mathcal{M}'$). The large overlapping between confidence bands $\mathcal{N}_{R_{ref}}^{\mathcal{M}}$ and $\mathcal{N}_{R_{ref}}^{\mathcal{M}'}$ shows that the diversity of generation is not sacrificed either.

## 5.2 IMPORTANCE SAMPLING

Under the same hypothesis of smooth manifold in section 5.1, the high-density region corresponding to the *worst-case* dense mode $I_m$ can be approximated with a convex hull.

### 5.2.1 METHOD DESCRIPTION

Importance sampling is a variance reduction strategy in the Monte Carlo method. Let the estimated neighboring function densities for the dense and sparse regions be $p_1$ and $p_2$ respectively. We accept the samples from $G$ falling in the high-density region with a probability of $p_2/p_1$, so that the calibrated densities can match.

We approximate the high-density region with a convex hull formed by the collection of latent codes $Z_{I_m}$ corresponding to the identities similar to $I_m$:

$$\text{Conv}(Z_{I_m}) = \{ \sum_{k=1}^{|Z_{I_m}|} \alpha_k z_k \mid (\forall k : \alpha_k \geq 0) \wedge \sum_{k=1}^{|Z_{I_m}|} \alpha_k = 1, z_k \in Z_{I_m} \} \tag{3}$$

### 5.2.2 EXPERIMENT

The experiment setting is mostly similar to the reshaping latent space via the Gaussian mixture models case. We integrate importance sampling to the latent code generation stage. Given the dense mode $I_m$, we can find the collection of latent codes $Z_{I_m}$ via sampling:

$$Z_{I_m} = \{z \mid d(I_m, G(z)) \leq d_0, z \sim N(0, \Sigma)\} \tag{4}$$

$Z_{I_m}$ is obtained from the top $10^2$ latent codes whose corresponding images have the smallest distances (1) to $I_m$, among the $10^6$ random samples. We randomly sample $10^6$ images from $\mathcal{M}$ and $\mathcal{M}'$ to form $\mathcal{T}$ and $\mathcal{T}'$ respectively. We plot the neighboring function curves for $I^M$ in $\mathcal{T}$ and $\mathcal{T}'$, and $I^{M'}$ in $\mathcal{T}$ and $\mathcal{T}'$ respectively. As shown in Figure 5, the mode collapse is again alleviated (indicated by a gap between the two red curves), without intensifying the clustering of $I'_m$ (indicated by a little gap between the two green curves), while FID only marginally increases from 5.93 ($\mathcal{M}$) to 5.94 ($\mathcal{M}'$). The confidence band $\mathcal{N}_{R_{ref}}^{\mathcal{M}}$ is overlapped with $\mathcal{N}_{R_{ref}}^{\mathcal{M}'}$, showing no loss of the diversity.

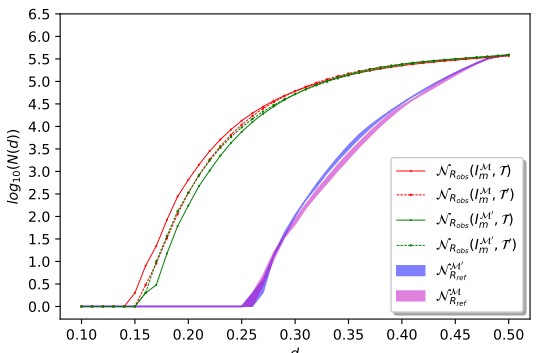

Figure 5: Identity clustering pattern analysis of Style-GAN on CelebA, before/after importance sampling.

Additionally, in the appendix, we show a *white-box* counterpart to the importance sampling approach, where the latent codes $Z_{I_m}$ are obtained via explicit optimization (accessing and altering model parameters). The *white-box* approach does not seem to notably outperform than our above black-box way, implying the relative effectiveness of the latter.

## 6 DISCUSSIONS AND FUTURE WORK

This paper is intended as a pilot study on exploring the mode collapse issue of GANs. Using face generation as a study subject, we quantify the general mode collapse via statistical tools, discuss and verify possible causes, as well as propose two black-box calibration approaches for the first time to alleviate the mode collapse. Despite the preliminary success, the current study remains to be limited in many ways. First, there are inevitably prediction errors for the identity descriptors from generated images, even we have performed our best efforts to find the three most accurate descriptor predictions. Moreover, the fundamental causes of GAN mode collapse demand deeper understandings. Besides, the two calibration approaches only handle one worst-case dense mode, leaving much improvement room open for future work.

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

# A APPENDIX

**A toy figure to explain our evaluation procedure**

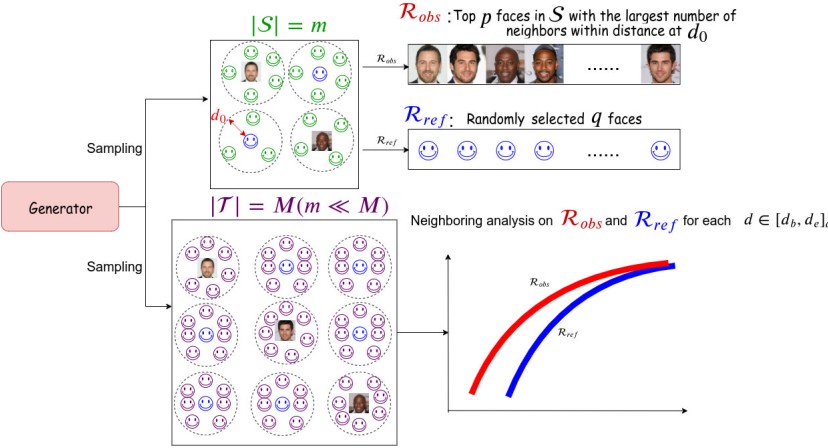

Figure 6: Illustration of our Algorithm 1: Identity Clustering Pattern Analysis via Sampling and Neighboring Function $\mathcal{N}$. See details in in Algorithm 1.

**Obtaining Latent Codes by Optimization (*White-box* Approach)** The second approach of finding $Z_{I_m}$ is latent code recovery via optimization:

$$Z_{I_m} = \{z \mid \min L_{vgg}(I_m, G(z)) + \gamma||I_m - G(z)||_2 + \alpha||z||_2, z \sim N(0, \Sigma)\} \tag{5}$$

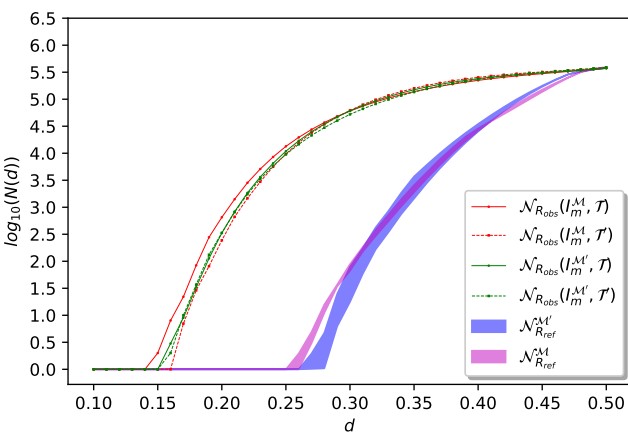

Figure 7: Clustering pattern analysis of StyleGAN on CelebA, before/after importance sampling.

Here we are using a combination of perceptual loss, $\ell_2$ loss and Gaussian prior. $\alpha$ and $\gamma$ are coefficients respectively. We run the optimization $10^2$ times with different random initialization of $z$ to get $10^2$ latent code. We then sample $10^6$ images from $\mathcal{M}$ and $\mathcal{M}'$ to form $\mathcal{T}$ and $\mathcal{T}'$ respectively. We plot the neighboring function curves for the *worst-case* dense mode $I_m$ in $\mathcal{T}$ and $\mathcal{T}'$, and $I'_m$ in $\mathcal{T}$ and $\mathcal{T}'$ respectively. As shown in Figure 7, the bias is again alleviated (indicated by a gap between the two red curves), while FID only marginally increases from 5.93 ($\mathcal{M}$) to 5.96 ($\mathcal{M}'$). The confidence band $\mathcal{N}_{R_{ref}}^{\mathcal{M}}$ is overlapped with $\mathcal{N}_{R_{ref}}^{\mathcal{M}'}$, showing no loss of the diversity.

We can observe that the FID score only marginally increases after white-box calibration. Meanwhile, the solid-red line (before calibration) is shifted to the right dash-red line (after calibration), indicating that the white-box approach can alleviate mode collapse problem and no severer mode is introduced. Besides, the solid-green line and dash-green line are almost overlaid, suggesting that the white-box approach has minimal impact on other modes.

**Evaluation results on different percentile instead of worst-case**    In the current evaluation metric, we only consider worst case because the worst case is the most important mode collapse problem we care in generative models, *i.e.*, if the model always generates similar faces, we can not use the generated dataset for other training purposes due to fairness/privacy concern. However, our evaluation metric can be easily adapted to detect median/mean dense regions by using faces with different percentiles instead of top k dense faces.

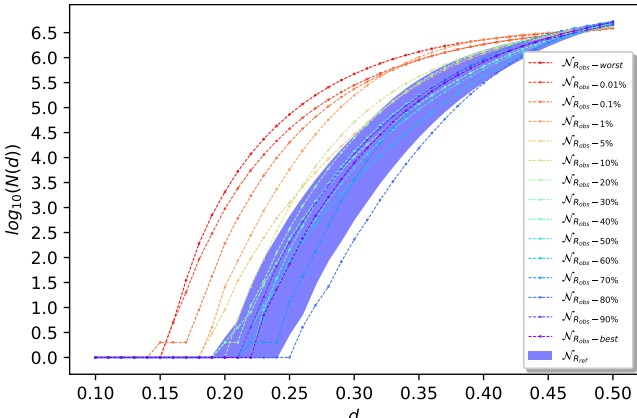

Figure 8: Identity clustering pattern analysis of StyleGAN on CelebAHQ using different percentile as clustering anchor

In this figure, we show the clustering pattern of choosing different percentiles in step 1 mode detection. We can observe that $R_{ref}$ overlays with $R_{obs} - 20\%$ and above, which demonstrats that around 20% of the generated images are highly clustered.

**Mode collapse between training data and testing data**    Our approach also have the capability to detect missing modes between training data and the generated data. If we simply change the first step to random sample m images from the training data, which could hopefully capture all modes in the training data, then we could find the missing mode by comparing the clustering pattern between training data and testing data.

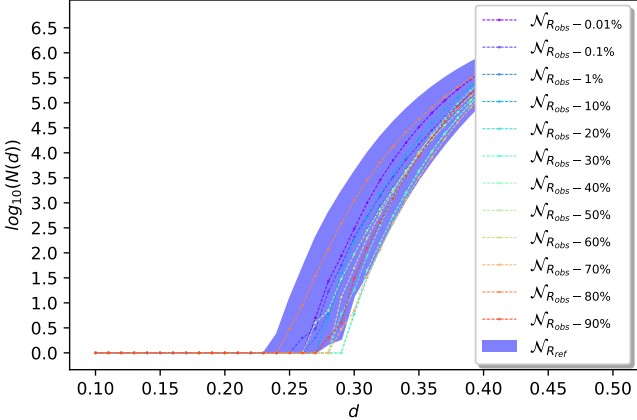

Figure 9: Identity clustering pattern analysis of StyleGAN on CelebAHQ using training data as clustering anchor

We can observe that the $R_{ref}$ overlays with $R_{obs}$ and we can not reject the hypothesis. Generated images are not severe clustered around anchors from training data, indicating that GAN models have the capability to generate "new faces". The width of the $R_{ref}$ band in Fig. 8 is similar to that in Fig. 9, suggesting that mode dropping is not severe between training data and generated images.

**Applying Our Proposed Metric on FFHQ**   FFHQ is a public face dataset contains 56,138 images, without repeating identities. We first randomly pick $1k$ images to form the $\mathcal{S}$ set and sort the $\mathcal{S}$ set according to the number of neighbors within distance 0.3. We choose the sample at percentile $0.01\%, 0.1\%, 1\%, 10\%, 20\%, 30\%, 40\%, 50\%, 60\%, 70\%, 80\%, 90\%$. We conduct the neighboring analysis on these selected samples. As is shown in Figure 10, we still observe a gap between $\mathcal{R}_{obs}$ and $\mathcal{R}_{ref}$, which demonstrates that FFHQ dataset has dense mode, even without repeating identities. Furthermore, we would like to clarify that our metric is proposed to measure the collapse of GAN's learned distribution. We have empirically shown in the paper that the mode collapse still occurs despite balanced training data.

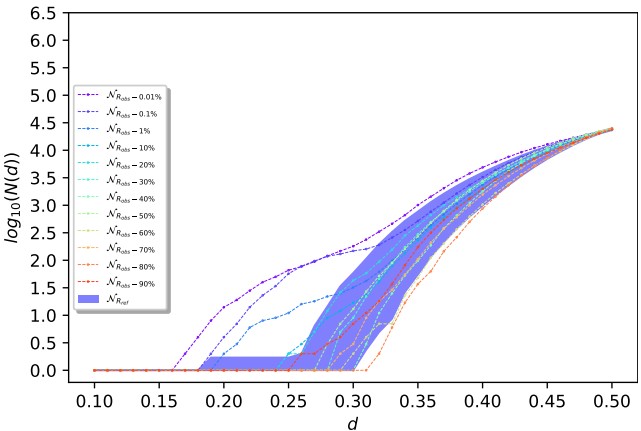

Figure 10: Identity clustering pattern analysis of StyleGAN on FFHQ.

