# OpenReview forum: "Is There Mode Collapse? A Case Study on Face Generation and Its Black-box Calibration"
_ICLR.cc/2020/Conference — Reject_

### Official Review · AnonReviewer1 · 2019-10-22
**Official Blind Review #1**

**Rating:** 6

**Review:**

This paper presents a set of statistical tools, that are applicable to quantitatively measuring the mode collapse of GANs. The authors consistently observe strong mode collapse on several state-of-the-art GANs using the proposed toolset. The authors analyze possible causes, and for the first time present two simple yet effective “black-box” methods to calibrate the GAN learned distribution, without accessing either model parameters or the original training data.

The writing and presentation are good.

My concerns regarding this paper are as below.
1) I wonder if the proposed method work for most GAN models, more experiments evaluated on more recent GAN-based  models should be added to verify the superiority claimed in this paper, e.g., TP-GAN [Huang et al., ICCV 2017], PIM [Zhao et al., CVPR 2018], DR-GAN [Tran et al., CVPR 2017], DA-GAN [Zhao et al., NIPS 2017], MH-Parser [Li et al., 2017], 3D-PIM [Zhao et al., IJCAI 2018], SimGAN [Shrivastava et al., CVPR 2016], AIM [Zhao et al., AAAI 2019].
2) The main contributions of this paper are not quite clear to me.
3) Typos need to be corrected in next version, e.g., all equations should have punctuation mark at the end, all e.g., i.e., et al., etc. should be italic, format of references should be consistent.

Based on my comments above, I decide to give the rate of WA for this paper.

**Experience Assessment:**

I have published in this field for several years.

**Review Assessment: Checking Correctness Of Derivations And Theory:**

I carefully checked the derivations and theory.

**Review Assessment: Checking Correctness Of Experiments:**

I carefully checked the experiments.

**Review Assessment: Thoroughness In Paper Reading:**

I read the paper thoroughly.

---

> ### Author Response · Authors · 2019-11-15
> **Response to Reviewer #1**
>
> Thank you for your careful reading and comments!
>
> Q1: We are evaluating the proposed metrics on more recent GAN-based models you suggested and will update the results once the results become available.
>
> Q2: The main contribution is listed as follows:
> 1. A pilot study of mode collapse existence in GAN.
> 2. Metric to detect mode collapse in GAN models without any labels (ground truth or pseudo-labels).
> 2. Black-box plug-and-play model collapse calibration.
>
> Thank you again for your careful reading and kindly identifying the typos in our paper! We will fix these typos and meticulously proofread our article.

---

### Official Review · AnonReviewer2 · 2019-10-23
**Official Blind Review #2**

**Rating:** 3

**Review:**

The direction of this work, to evaluate whether mode collapse exists without using label information, is very good. Rather than using the labels, the authors use an off-the-shelf model (on faces) to provide a space on which to measure distances between generated images. They use this distance to test the hypothesis that samples are highly concentrated around some modes, thus under representing others.

The paper could benefit from more clarity. For instance, the methods would be better illustrated through some toy figures. In addition, some explanations in this work are very hard to parse, e.g., the first paragraph of the methods section.

While this work is focusing on black box methods for evaluating and palliating mode dropping (aka collapse), it's a bit disappointing that these results are at least also evaluating on white-box type methods in settings where mode dropping is clearer, e.g. PACGAN on stacked MNIST or even normal CelebA. Unfortunately, no white-box methods are covered in this work, so there is no strong point of comparison, which would be helpful to establish the validity of this work.

Finally, the authors demonstrated that there exist a high-density mode, but not whether some modes might be missing. How can this method be used to find missing modes if the generator isn't generating them without the real data? If my generator is only generating a few digits, but each of them represented similarly by the generated distribution, what would this measure do? It wont detect that the generator is missing modes: you'd need to know those modes existed (e.g., have examples of them).

Generally I like the experiments, though I wish there were more qualitative results looking at more than just the existence of one worst-case mode.

Other comments
page 1
I'm not sure the connection between mode collapse and instability is well-established. What motivates connecting to instability in this work?
The statement about co-variate shift is a little vague, and it's not clear what the connection to mode collapse is.
page 2
I'm surprised PACGAN isn't mentioned in the white-box methods. It was one of the big SOTA methods for palliating mode dropping.
I'm not sure why ignore white-box methods: at least it would be good verification that this method works (e.g., across methods, common measures of collapse used in those settings, etc)
page 3
The first paragraph of Section 3 is very difficult to understand.
f is normalized?
page 4
Why not use Ripley's K? (this is not explained, and should be)
The main link missing in the test proposed is that to mode dropping. The problem is that this measure wont detect mode dropping if there's aren't samples from those modes to measure anything against. You need real samples as well.
How were these face models chosen? Why not use the discriminator of the GAN (at least test what it does)? How were these hyper parameters chosen?
page 7:
5.1.2 could use an accompanying toy figure demonstrating what's going on.

---- Update ---
Thanks for your responses and the updated experiment on the white-box calibration. I still think that comparisons to existing methods that palliate mode-dropping is essential for this work, and I would encourage you to bring experiments that allow for comparison to PACGAN et al into the main text for future versions, as the work / direction as value. I will keep my score as-is.

**Experience Assessment:**

I have published one or two papers in this area.

**Review Assessment: Checking Correctness Of Derivations And Theory:**

I assessed the sensibility of the derivations and theory.

**Review Assessment: Checking Correctness Of Experiments:**

I assessed the sensibility of the experiments.

**Review Assessment: Thoroughness In Paper Reading:**

I read the paper at least twice and used my best judgement in assessing the paper.

---

> ### Author Response · Authors · 2019-11-15
> **Response to Reviewer #2**
>
> Thank you for your careful reading and comments!
>
> - Q1:  Thanks for kindly pointing out our writing issues. We add a toy figure to better explain our evaluation process, please see Appendix Figure 6 in the revised paper.
>
> - Q2: We basically agree with your comment that evaluating on white-box type GAN mode-collapse calibrations need to be included. We attach the results of a white-box approach, i.e., importance sampling, in Appendix Figure 7. We are conducting the experiments on PacGAN that you have suggested and will update the results in comments once they become available, and also add them to the paper’s next update.
>
> We can observe that the FID score only marginally increases after the white-box calibration. Meanwhile, the solid-red line (before calibration) is shifted to the right dash-red line (after calibration), indicating that the white-box approach can alleviate the mode collapse problem and no severer mode is introduced. Besides, the solid-green line and dash-green line are almost overlaid, suggesting that the white-box approach has minimal impact on other modes.
>
> - Q3: In the current evaluation metric, we only consider worst case because the worst case is the most important mode collapse problem we care in generative models, i.e., if the model always generates similar faces, we can not use the generated dataset for other training purposes due to fairness/privacy concern.
>
> However, our evaluation metric can be easily adapted to detect median/mean dense regions by using faces with different percentiles instead of top k dense faces (please see Appendix Fig. 8 in the revised paper). In this figure, we show the clustering pattern of choosing different percentiles in step 1 mode detection. We can observe that R_ref overlays with R_obs-20% and above, which demonstrates that around 20% of the generated images are highly clustered.
>
> Besides, our approach also has the capability to detect missing modes between training data and the generated data by estimating the probability of any specific face mode being sampled from the generator. Given two images $I_0$ and $I_1$, we define similarity between $I_0$ and $I_1$ as $1-d(I_0, I_1)$, where the distance $d$ follow the same definition of equation 1 in the paper. The expected similarity between $I_0$ and a random sampled face $G(z)$ is defined as $\int s(I_0, G(z))p(z)dz$, where $z \sim N(0, \Sigma)$, $I=G(z)$. The integral can be computed more efficiently by importance sampling on $z$, such that $G(z)$ is closer to $I_0$ in distance.
>
> We can detect mode dropping by comparing the expected similarity of the dropped mode to a randomly sampled face, with those non-dropped mode's similarity to a randomly sampled face. The smaller the expected similarity is, the more strongly the mode is dropped.
>
> Moreover, if we simply change the first step to randomly sample m images from the training data to capture enough modes, then we could find the missing modes by comparing the clustering pattern between the training data and the testing data. (please see Appendix Fig. 9 in the revised paper). We can observe that the R_ref overlays with R_obs and we can not reject the hypothesis. Generated images are not severely clustered around anchors from the training data, indicating that GAN models have the capability to generate "new faces". The width of the R_ref band in Fig. 7 is similar to that in Fig.8, suggesting that mode dropping is not severe between the training data and the generated images.

---

> ### Author Response · Authors · 2019-11-15
> **Response to Reviewer #2 (Continued)**
>
> Q1 Connection between mode collapse and instability:
> The stability is more related to the GAN model itself, but we care more about the diversity of GAN generated results. We tried to alleviate the mode collapse problem while maintaining the stability of GAN models.
>
> Q2 Connection between co-variate shift and mode collapse is:
> We followed the notion in this paper (https://arxiv.org/pdf/1711.00970.pdf) and consider mode collapse as a type of covariate shift: “We demonstrate two specific forms of covariate shift caused by GANs: 1) Mode collapse, which has been observed in prior work (Goodfellow, 2016; Metz et al., 2016); 2) Boundary distortion, a phenomenon identified in this work and corresponding to a drop in diversity of the periphery of the learned distribution.”
>
> Q3 PACGAN is not mentioned in the white-box methods
> We agree with your comment that evaluating on white-box type GAN mode-collapse calibrations need to be included. We attach the results of a white-box approach, i.e., importance sampling, in Appendix Figure 7. We will definitely include PACGAN results in the revised version once they become available.
>
> Q5 Why not use Ripley's K?
> The Ripley's K function compares the expected number of points within a local neighborhood of radius r at any point in the dataset versus the expected density assuming complete spatial randomness (CSR). Our proposed neighboring function N is a surrogate of the K function. There are two reasons why we cannot use the Ripley's K.
> Firstly, there is neither spatial coordinates for each face sample nor a specified spatial region in the embedded face feature space. We can only get a pairwise distance between two face samples. Secondly, there is no complete spatial randomness (CSR) in the embedded face feature space. The standard model for CSR is that events follow a homogeneous Poisson process over the study area, which does not hold in the embedded face feature space.
>
> Q7 How were these face models chosen?
> We choose the off-shelf face embedding models i.e. SOTA face recognition model. In fact, we could use any feature engineering/dimension reduction approaches such as PCA, sparse reprojection, neural network here. But we choose to use face recognition feature embedding here for three reasons:
> a) Those features capture most information (rich semantics, strong transferability) since we can recognize a face identity, gender, race, etc. just based on those features.
> b) Traditional feature engineering/dimension reduction approaches are not as efficient, e.g. PCA can not scale up.
> c) The feature embedding is more general, transferable and is independent of the data and model used in GAN training.
>
> Q6 mode dropping detection
> Please see Q3 in our last response.
>
> Q4 & Q8 toy figure to demonstrate evaluating procedure
> Please see Appendix Figure 6 in the revised paper, which could better clarify our evaluating procedure. Yes, the features are normalized.

---

### Official Review · AnonReviewer3 · 2019-11-04
**Official Blind Review #3**

**Rating:** 1

**Review:**

This work addresses the important problem of generation bias and a lack of diversity in generative models, which is often called model collapse. It proposed a new metric to measure the diversity of the generative model's "worst" outputs based on the sample clustering patterns. Furthermore, it proposed two blackbox approaches to increasing the model diversity through resampling the latent z. Unlike most existing works that address the model collapse problem, a blackbox approach does not make assumptions about having access to model weights or the artifacts produced during model training, making it more widely applicable than the white-box approaches.
In terms of experiment setup, the authors chooses face generation as the area to investigate and measures the diversity by detecting the generated face identity. With the proposed methods, the authors showed that most STOA methods have a wide gap between the top p faces of the most popular face identities and randomly sampled faces. It further showed that the proposed blackbox approaches increases the proposed diversity metric without sacrificing image quality.

The proposed diversity measuring metric is lacking both in terms of experimental proofs and intuitive motivations. While the black-box calibration of a GAN model may be attractive under specific settings, the authors did not consider the restrictions under those situations and their design may be hard to implement as a result. For those reasons, I propose to REJECT this paper.

Missing key experiments that will provide more motivation that 1. the new metric reflects human perception of diversity 2. the new metric works better than existing ones:
1. Please provide experiments and/or citation for using the face identity as a proxy for face image diversity. this is important since all your experiments rely on that assumption.
2. Were there experiments that applies your metric to the training datasets like CelebA and FFHQ? In theory your metric should show no gap between N_R_obs and N_R_ref measured on the training dataset since that's the sampled ground truth.

Missing assumptions about blackbox calibration approaches:
1. If we do not have access to the model parameter, the training data, or the artifacts during training like the discriminator, what are some of the real world situations that fit this description? In those cases, is it too much to assume that we can control the random seed input to G?
2. Is it reasonable to assume some constraints on how much data we can get from the blackbox generator? A website that just exposes the image generation API may not allow you to ping their service 100k times to improve the generation diversity. If you are allowed to do that, it may be reasonable to assume that you can contact the API provider to get access to the rest of the model.

Minor improvements that did not have a huge impact on the score
1. I found the argument about FID in section 2.1 unconvincing. Are there proofs or citations for the claim that real images don't follow multivariate gaussian distribution after applying FID? Copying is indeed an issue that FID cannot detect, but it may be tangential to model collapse for real world concerns like privacy.
2. The statement "IS, FID and MODE score takes both visual fidelity and diversity into account." under "Evaluation of Mode Collapse" is contradictory to the description in sec 2.1 that IS in fact does not measure diversity.
3. You may want to consider stating the work as "a pilot study" (sec 6.) earlier in the abstract or in the introduction, so that the reader knows what to expect.


**Experience Assessment:**

I have read many papers in this area.

**Review Assessment: Checking Correctness Of Derivations And Theory:**

I carefully checked the derivations and theory.

**Review Assessment: Checking Correctness Of Experiments:**

I assessed the sensibility of the experiments.

**Review Assessment: Thoroughness In Paper Reading:**

I read the paper at least twice and used my best judgement in assessing the paper.

---

> ### Author Response · Authors · 2019-11-15
> **Response to Review #3**
>
> Thank you for your careful reading and comments.
>
> Q1: Missing assumptions about the black-box calibration approaches
> We thank R1 and R2 for endorsing the merit of our proposed black-box calibration.
>
> The black-box calibration assumes no read/write to model weights or availability training data, but access to the sampling of random seed. The black-box calibration is useful for both model user and API owner.
> Model owner: We suppose that the dense mode happens to be close to a specific training image, thus violating privacy. The model owner would like to calibrate the model to alleviate the mode collapse. In such a situation, training data may no longer be accessible since it contains private information, e.g. human faces or person images. Retraining consumes much time and energy, especially for complex models trained on a huge dataset. Besides, we empirically validate that the dense mode is not caused by imbalanced data or randomness during initialization/optimization. So retraining won't work for dense-mode alleviation. Our proposed black-box calibration has an advantage over retraining with minimum time and energy cost and no touching training data. Moreover, the calibration can target any dense mode for alleviation.
> API owner: For enterprise users having access to the face image generation service via cloud API, they are given the ping service for a huge number of times or not even restricted. Black-box calibration enables the API owner to customize the model's sampling process to meet the users' needs.
>
>
> Q2: Missing key experiments that will provide more motivation that 1. face identity can be used as a proxy for face image diversity; 2. applying our proposed metric to the training datasets should show no gap between $\mathcal{R}_{obs}$ and $\mathcal{R}_{ref}$:
>
> 1. Face identity as a proxy for face image diversity
>
> We would like to clarify that we are not using the identity label as a proxy. Instead, we are using the embedding features obtained from the neural network trained on the face recognition task. We claim that the embedding features have rich semantics of all kinds of facial attributes, e.g. age, gender, race and so on. The rich semantics of the face embedding feature can be validated by its strong transferability on other visual tasks, e.g. gender/race classification and age regression. Prior studies [Savchenko, Andrey V, "Efficient facial representations for age, gender and identity recognition in organizing photo albums using multi-output ConvNet" (2019)] have shown that transfer learning using neural networks pretrained on face recognition can produce highly effective results for gender recognition and age estimation.
>
> 2. Applying our metric on the training set of FFHQ
>
> FFHQ is a public face dataset contains $56,138$ images, without repeating identities. We first randomly pick $1k$ images to form the S set and sort the S set according to the number of neighbors within distance 0.3. We choose the sample at percentile $0.01\%, 0.1\%, 1\%, 10\%, 20\%, 30\%, 40\%, 50\%, 60\%, 70\%, 80\%, 90\%$. We conduct the neighboring analysis on these selected samples. We still observe a gap between $\mathcal{R}_{obs}$ and $\mathcal{R}_{ref}$, which demonstrates that FFHQ dataset has dense mode, even without repeating identities. Furthermore, we would like to clarify that our metric is proposed to measure the collapse of GAN's learned distribution. We have empirically shown in the paper that the mode collapse still occurs despite balanced training data. You can check the details in the appendix of the paper, the paragraph of "Applying Our Proposed Metric on FFHQ".
>
> Q3: Minor improvements
>
> 1. Proof or citation for the flaws of FID
> There is a recently published survey paper that can back our claim. It is [Ali Borji, "Pros and Cons of GAN Evaluation Measures" (Arxiv 18)]
>
> 2. The contradiction between the two statements
> We use the word "loss of diversity" since IS's measuring of diversity is limited. E.g., on ImageNet with 1000 classes, it can not rule out the case when then generator simply repeating the same image for each different class.
>
> 3. We take your advice and will address this piece of work as a "pilot study" in the final version.

---

### Decision · Program_Chairs · 2019-12-19

**Decision:**

Reject

**Comment:**

This paper studies the problem of mode collapse in GANs. The authors present new metrics to judge the model's diversity of the generated faces. The authors present two black-box approaches to increasing the model diversity. The benefit of using a black box approach is that the method does not require access to the weights of the model and hence it is more easily usable than white-box approaches. However, there are significant evaluation problems and lack of theoretical and empirical motivation on why the methods proposed by the paper are good. The reviewers have not changed their score after having read the response and there is still some gaps in evaluation which can be improved in the paper. Thus, I'm recommending a Rejection.